# UAS Traffic Management Communications: The Legacy of ADS-B, New Establishment of Remote ID, or Leverage of ADS-B-Like Systems?

**Neno Ruseno [1,\*][ID], Chung-Yan Lin [2] and Shih-Cheng Chang [3]**

1   Department of Power Mechanical Engineering, National Formosa University, Yunlin 632301, Taiwan
2   Department of Aeronautical Engineering, National Formosa University, Yunlin 632301, Taiwan; frank.lin@nfu.edu.tw
3   Department of Aeronautical Engineering, Chaoyang University of Technology, Taichung City 413310, Taiwan; serchen@cyut.edu.tw
*   Correspondence: d0975107@gm.nfu.edu.tw

**Abstract:** Unmanned aerial system (UAS) traffic management (UTM) requires each UAS to communicate with each other and to other stakeholders involved in the operation. In practice, there are two types of wireless communication systems established in the UAS community: automatic dependent surveillance broadcast (ADS-B) and remote identification (Remote ID). In between these two systems, there is ADS-B-like communication which leverages using other types of communications available in the market for the purpose of UTM. This review aims to provide an insight into those three systems, based on the published standard documents and latest research development. It also suggests how to construct a feasible communication architecture. The integrative approach is used in this literature review. The review categorization includes definition, data format, technology used, and research applications, and any remaining issues are discussed. The similarities and differences of each system are elaborated, covering practical findings. In addition, the SWOT analysis is conducted based on the findings. Lastly, multi-channel communication for UTM is proposed as a feasible solution in the UTM operation.

**Keywords:** UTM communication; ADS-B; ADS-B-like communication; Remote ID; integrative approach; SWOT analysis; multi-channel communication



## 1. Introduction

The latest unmanned aerial vehicle (UAV) could be used for photography, agriculture, surveillance, transportation, etc. Those UAV applications could be categorized into defence, enterprise, consumer, public safety, logistics and passenger. According to a white paper from Levitate Capital, it is predicted that the highest market size growth is in logistics, and it increased by more than 300 times from less than USD 0.1 billion (in the year 2020) to USD 33 billion (in the year 2030). Further forecasts mention that the Asia Pacific region will experience the highest growth, from USD 4 billion (in the year 2020) to USD 30 billion (in the year 2030) [1].

The growth of UAVs will make the sky more crowded. Hence, unmanned aerial system (UAS) traffic management (UTM) is required to organize the operation of the UAS to ensure its safety and efficiency for the public and all its stakeholders. Moreover, it is a supplement to the air traffic management (ATM) of manned flights, which shares the same airspace region.

According to a research and market consultant report, the UTM system's market value is forecasted to grow by 17.13% in the next ten years. It is led by regions of North America and Europe, where the development of UTM systems has reached the trial and demonstration stage and is fully supported by the authority and the industries [2].

A UTM system requires that each UAV be able to see and be seen; therefore, the communication channel becomes crucial. In the last five years, state-of-the-art research has focused on three types of communication: ADS-B, ADS-B-like communication, and Remote ID. ADS-B is the legacy system currently used in manned flights and has been mandatory since 2020. Similarly, the ADS-B-like system works similarly to ADS-B but uses different types of wireless communication technology such as 4G, XBee, and APRS (automatic packet reporting system). However, Remote ID is a new standard established, based on Bluetooth/Wi-Fi technology that fulfills the requirement for current UTM operations. Since there is no obligation regarding which technology should be adopted, each country must understand each of those systems used in the UTM system. Hence, a critical review of those technologies is essential.

This research aims to provide insight into those three communication systems, based on the published standard documents and latest research development. The novelty of this review is its inclusion of the range of communication links available, including the new Remote ID protocol, and suggestions for how to construct a feasible communication architecture. Based on the review, a new UTM communication architecture will be proposed. The assumed environment for UAS operation is at a low altitude (below 400 ft), with a lightweight object (less than 50 kg) and at a short-range (less than a 1 h flight). The manuscript consists of a review methodology in Section 2, the result of the review in Section 3, and a discussion in Section 4. The last section is a conclusion and recommendation.

## 2. Review Methodology

The methodologies in conducting a literature review can be classified into three approaches: systematic, semi-systematic, and integrative. Typically, the systematic approach aims to synthesize and evidence the comparison. It uses quantitative analysis such as the statistical method. The next is the semi-systematic approach, which aims to overview the research area and timeline development. It normally uses a quantitative or qualitative analysis method. However, the integrative approach aims to critique and synthesize. It uses a qualitative analysis method based on research articles, books, and other published texts [3].

The authors believe that the integrative approach is appropriate for our purpose in reviewing UTM communication. Since UTM communication combines old and new systems, it needs to be critiqued and synthesized. Our analysis will be based on the published research articles and standard system documents for that purpose. In addition, a qualitative analysis method is selected to review all the available literature. In this integrative approach, the literature review process consists of four phases: design, conduct, analysis, and writing [4], as shown in Figure 1.

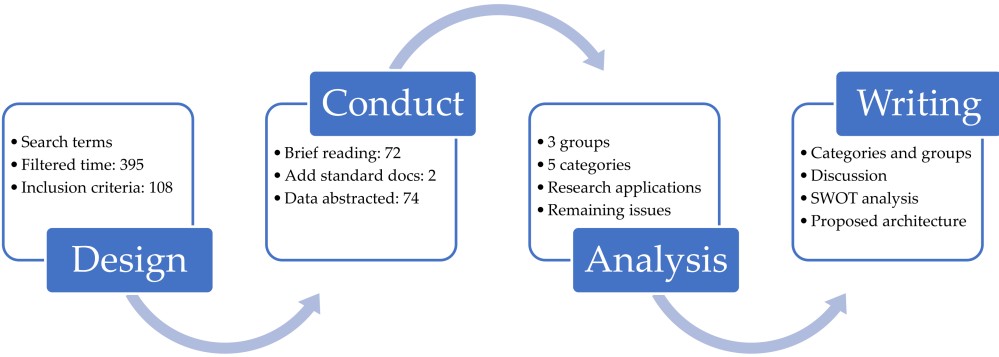

**Figure 1.** Literature review process.

### 2.1. Designing the Review

The first step in designing a literature review is defining the search terms used. Based on our topic research, the selected search terms were:

- "ADS-B" and (UAS, UAV, or UTM)

- "Remote ID" and (UAS, UAV, or UTM)
- (UAS or UAV) and (Communication or Cellular)

The search terms were used in several publication websites to search the related literatures based on their titles and keywords. We also filtered out the literatures which were published before 2017. The result was 395 literatures, of which the first and second terms had 49 and 31 results, and the third term resulted in 315 data.

Then, inclusion criteria were applied to the literatures which focus on UTM communication, were written in English, and were published in a conference proceeding or journal. The inclusion process resulted in 108 literatures selected for further analysis.

### 2.2. Conducting the Review

The selected literature from the previous process was read briefly to define its content related to UTM communication purposes. The process resulted in 72 literatures being selected as the final list. Many literatures were unselected because they emphasized using a UAS as a means to support communication systems, which is not related to our topic. Since ADS-B and Remote ID are the regulated communication systems in aviation, the authority has published standard documents. Thus, we added two standard documents for ADS-B and Remote ID to our list of literatures.

The literatures in the final list were read in-depth, especially in the method and result sections, to abstract the data. The abstraction data were used to fulfill the purpose of the review and the analysis type conducted.

### 2.3. Qualitative Analysis

Based on the abstracted data, we divided the literatures into three groups, which were ADS-B, Remote ID, and ADS-B-like communication. For each group, they were categorized into five categories, which were definition, data format, technology, research application, and remaining issues, as shown in Table 1. The definition, data format, and technology categories are used to give the readers an understanding of all the systems and how they work. The research applications and remaining issues categories aim to review the latest research development and future trends.

**Table 1.** List of documents and their categories.

| No | Category | ADS-B | ADS-B-Like | Remote ID |
|----|----------|-------|------------|-----------|
| 1 | Definition | [4–7] | [8–11] | [12–14] |
| 2 | Data format | [5,7,15] | [8,16,17] | [13,18,19] |
| 3 | Technology | [5–7,15,20,21] | [8–11,16,17,22,23] | [12–14,18,19] |
| 4 | Research application | [4–6,20–22,24–52] | [8,11,16,17,22,41,43,45,53–62] | [12–14,19,42,63–68] |
| 5 | Remaining issues | [69–74] | [10,69,70,73–76] | [70,72–74] |

Since there are many types of research applications and the remaining issues are highlighted in the literatures, these categories are broken down further into several types. The research applications consist of surveillance, detect and avoid, capacity estimation, communication, command and control, and security categories, as shown in Table 2. Whereas, the remaining issues consist of security, safety, communication, and surveillance, as shown in Table 3.

### 2.4. Structure and Writing the Review

The structure of the literature review is based on the categorization in Tables 1–3. The writing of the review results covers all categories, and qualitative analysis is used in the discussion.

**Table 2.** List of documents in research application.

| No | Research Application | ADS-B | ADS-B-Like | Remote ID |
|----|---------------------|-------|-----------|-----------|
| 1 | Surveillance | [24–33] | [8,11,16,17,53–57] | - |
| 2 | Detect and avoid | [4,21,22,34–49] | [22,41,43,58–60] | [12–14] |
| 3 | Capacity estimation | [6,20] | - | [63] |
| 4 | Communication | - | [45,61] | [19,64] |
| 5 | Command and control | - | [53,57] | - |
| 6 | Security | [5,50–52] | [62] | [42,65–68] |

**Table 3.** List of documents in remaining issue.

| No | Remaining Issues | ADS-B | ADS-B-Like | Remote ID |
|----|-----------------|-------|-----------|-----------|
| 1 | Security | [69–72] | [70–75] | [72–74] |
| 2 | Safety | [69–71] | [70,73] | [12–14] |
| 3 | Communication | - | [69,73,74,76] | - |
| 4 | Surveillance | [73,74] | [10,69] | [70] |

## 3. Result of Review

This literature review covers ADS-B, Remote ID, and ADS-B-like systems as the UTM communication systems. The writing consists of five categories: definition, data format, technology, research application, and remaining issue.

### 3.1. Definition

ADS-B is an abbreviation of automatic dependent surveillance broadcast. It is a surveillance technology in aviation that automatically and periodically broadcasts its flight information. The broadcasted data include altitude, heading, and position, dependent on the global position system (GPS) [4]. The position information is generally based on a global navigation satellite system (GNSS). It is not only GPS, but also GLONASS, COMPAS, or GALILEO systems [5]. The ADS-B system framework broadcasts information between sender and receiver, as shown in Figure 2. The information broadcast is in the ADS-B frame, which can be received by other airborne aircrafts or nearby ground systems [7]. The main function of the ADS-B is a surveillance application conducted by air traffic control (ATC) to monitor the flights in its airspace [6].

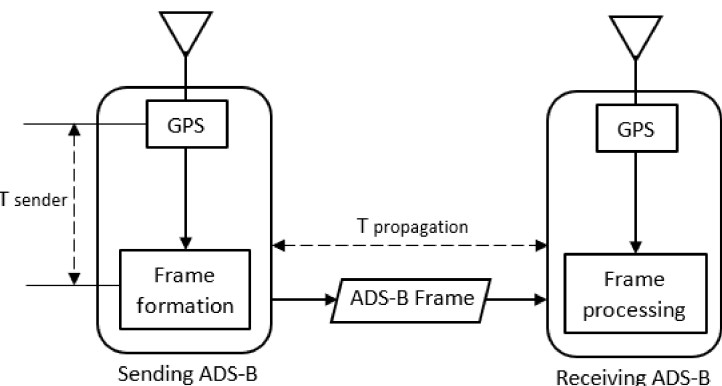

**Figure 2.** ADS-B broadcasting framework [5].

Besides ADS-B, the new communication system in the UAS is remote identification (Remote ID). It functions as a license platform that transmits information in the form of a REST API, including the owner of the UAV and their location. The information can be accessed by the authorities [12]. Its main purpose is security [14], as illustrated in Figure 3.

The security personnel can receive a direct signal from the UAVs via their device for security monitoring. Moreover, the UAV information can also be received by other stakeholders such as the public in nearby locations [13].

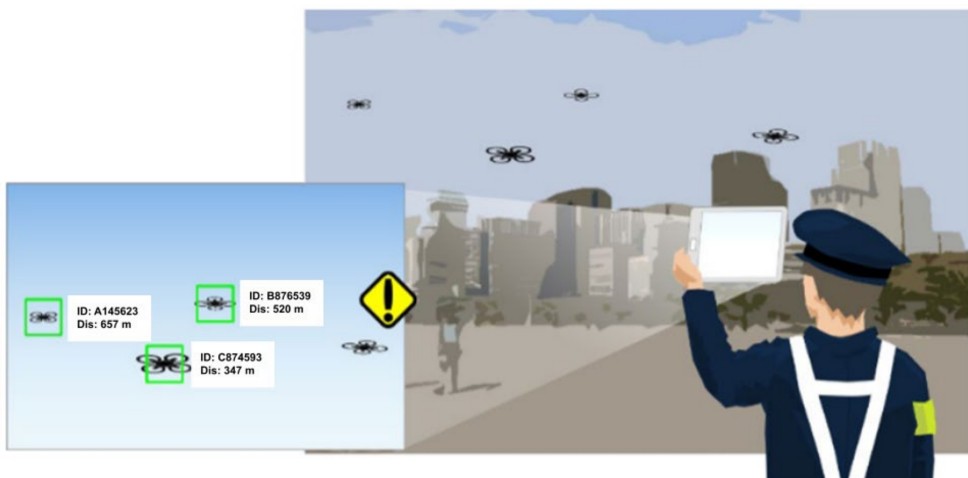

**Figure 3.** Illustration of Remote ID application in public security [14].

The last system is called ADS-B-like communication. It is a surveillance system that is similar to the ADS-B system but uses other technologies available in the market, such as long-term evolution (LTE) networks [8], the same spectrum with the upload (UL) of cellular ground users (GUEs) [9], cellular architectures (5G UAVs) [10], and broadband technologies such as Wi-Fi beacons, APRS, XBee, and LoRaWAN (long-range wide area network) [11]. One of the UTM concepts, which consists of an ADS-B-like communication system, is shown in Figure 4. It combines 4G/LTE, XBee, APRS, and LoRaWAN [16].

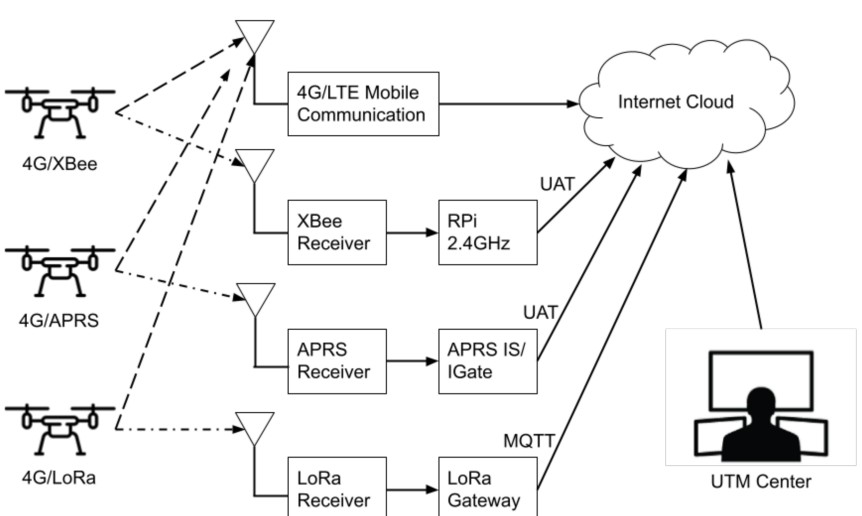

**Figure 4.** UTM based ADS-B-like communication architecture [16].

### 3.2. Data Format

ADS-B messaging consists of 112 bits, including downlink format, capability, aircraft address, ADS-B data, and parity check, as shown in Figure 5. The ADS-B data are represented in 56 bits, containing various types of messages. The first five bits in ADS-B data are the type of code (TC) field, which identify the message type. These types range from routine messages, such as identification and positioning, to specific and indirect messages, such as status and target state [5,7].

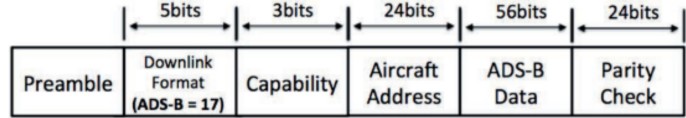

**Figure 5.** ADS-B message protocol [5].

In the ICAO document, the ADS-B data consist of Asterix Cat 21 across group 1 (mandatory), group 2 (desirable), and group 3 (optional). They must broadcast all operational data (group 2) and the required data (group 1) in Asterix messages. However, group 3 (optional data) depends on specific operational needs [15]. The ADS-B protocol in UAV usage is the same as in manned flights. Only the size and weight matter and need to be adjusted to suit the UAV operational condition.

Since ADS-B is not mainly for UAVs, another means of communication is introduced: the Remote ID. Its main purpose is for security reasons. When a security officer observes a UAV flying overhead, as shown in Figure 6, they can observe the Remote ID signal and verify it via REST API protocol to the UTM system. The detailed information that can be obtained is [19]:

- UAS owner identification and its contact.
- Properties of the vehicle including aircraft type, such as fixed-wing, quadcopter, etc.
- Current flight plan, vehicle heading, speed, and future operations.
- UTM status of the current flight plan, such as rogue, non-conforming, etc.

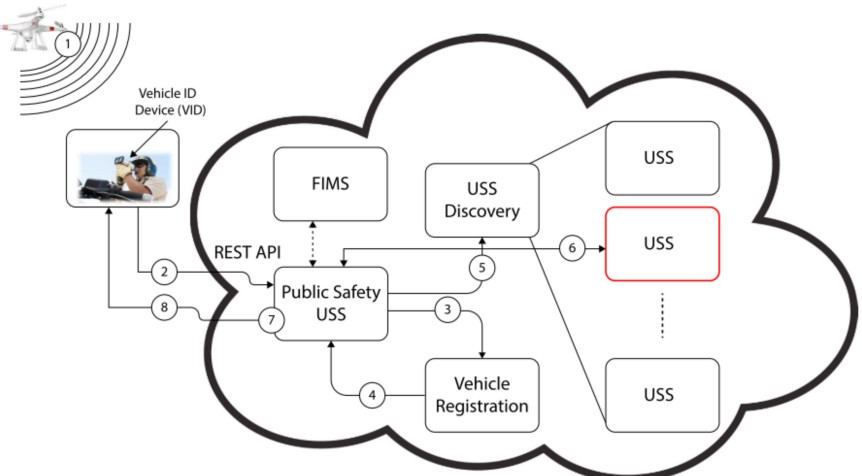

**Figure 6.** Remote ID flow process diagram [19].

Moreover, Remote ID could also function as vehicle-to-vehicle (V2V) communication, which broadcasts information containing vehicle information, position (latitude, longitude, and altitude), time, distress status, sent count, and source [13]. The complete required and optional data fields for Remote ID are given in the document published by ASTM. It describes the minimum characteristics that must be supported by both network and broadcast implementations [18].

Like the ADS-B system, the ADS-B-like system can also contain surveillance data, which are designed by 90 bytes to include heading (5 bytes), UAV (6 bytes), pilot (6 bytes), latitude (9 bytes), longitude (10 bytes), altitude (4 bytes), 6 degrees of freedom (p, q, r, u, v, and w) (36 bytes), velocity (6 bytes), acceleration (6 bytes), and tail (2 bytes) [16,17]. Since the advent of UAS technology, the broadcasted information can be completed by the control station location, future aircraft intent information, mission intention, degraded states, and pilot in command information [8]. However, the ADS-B-like system could employ different communication technologies, such as cellular networks and Wi-Fi.

*3.3. Technology*

ADS-B uses high radio frequency to transmit its message. There are two types of ADS-B broadcasting frequencies, which are a 978 MHz universal access transceiver (UAT) [20] and a 1090 MHz extended squitter (ES). The ADS-B message's standard mean sending rate is 6.2 messages/second, and the message duration is 120 μs [6]. In comparison, the message duration for the secondary surveillance radar (SSR) messages is 35 μs, and mode S short messages are 64 μs [6]. The ADS-B message has a slightly longer required transmission duration.

In manned airplanes, the power required for ADS-B ranges from 75 W to 500 W, with the maximum broadcasting range up to 500 km. While the ADS-B power on UAVs is much lower, it is between 0.01 W and 1 W [6]. Generally, ADS-B modules consist of two functions, which are ADS-B OUT (transmitter) and ADS-B IN (receiver). ADS-B IN facilitates reception and demodulation of nearby ADS-B OUT broadcasts, while ADS-B OUT periodically broadcasts information regarding its states [7]. Hence, the surveillance system consists of these broadcaster and receiver capabilities.

An example of an ADS-B module for UAV are the Ping2020 ADS-B transponders produced by uAvionix. It can transmit identity, aircraft position, heading, and speed information extracted from a flight controller. This ADS-B transponder broadcasts data in every second, and its barometric altitude sensor has a reading accuracy of up to 10 cm [21].

Unlike ADS-B, Remote ID can use several communication technologies to transmit its messages, such as LoRaWAN, infra-red-light beacons, dedicated short-range communication (DSRC) radios, and Bluetooth 5 low energy (BLE5). Firstly, LoRaWAN is an affordable, long-range communication of the internet. It has a coverage range of 15 to 30 km [12]. Secondly, an infra-red light beacon is used to provide a vehicle registration and model database (VRMD), named UTM vehicle identification number (UVIN) [19].

The third, DSRC radios, can be used as V2V communication devices in a Remote ID system with a frequency band from 5.850 to 5.925 GHz and a +20 dBm power output. They assign for each UAS a globally unique flight identifier (GUFI) as the Remote ID identity [13]. Last is BLE5, which is a low-power, unlicensed Bluetooth radio frequency. According to the flight test between a small UAS and a manned helicopter using BLE5, the results indicate that the possible performance range is about 1 km [14].

Furthermore, based on the Remote ID standard published by ASTM, there are two types of Remote ID. They are network Remote ID and broadcast Remote ID. Network Remote ID employs a cellular network to transmit its message. At the same time, broadcast Remote ID employs the short-range wireless technology mentioned earlier [18]. The broadcast Remote ID is endorsed by FAA and will be effective by 2023. However, the Network Remote ID is not yet regulated due to some rejections from the public opinion.

Like Remote ID, the ADS-B-like system employs several communication technologies to transmit its message. There are two groups of wireless technology used in the ADS-B-like system: cellular-based and non-cellular-based technology. The cellular-based technology includes LTE/4G networks with frequency spectrum at 979 MHz or 1104 MHz [8], and 5th generation (5G) with a frequency spectrum from 24 GHz to 86 GHz [9] [10]. The non-cellular-based technology consists of LoRaWAN [16], XBee [22], APRS [17], and Wi-Fi beacons [11]. LoRaWAN uses a technique of spread spectrum modulation. It is an unlicensed radio spectrum ranging from 920 MHz to 925 MHz, and is used in industrial, scientific, and medical applications [16]. The second is XBee, which uses the Digi Mesh protocol at 900 MHz. It uses a low power of 1W/32dBm and ranges from 9 to 65 miles [16]. The third is APRS, which is commonly used by volunteer rescue groups with a specific frequency of 144.64 MHz (assigned for public use) [17]. The last is Wi-Fi beacons, which use the 2.4 GHz band and 5 GHz band. They could achieve an output of at least two messages/second in the worst scenario using only 17 dBm, which equals about 50 mW of power transmission [11].

Unlike ADS-B and Remote ID, which have fixed standards and are regulated by the authority, the ADS-B-like system is an open system suggested by the research community

that leverages the available wireless technologies. When we compare Remote ID and ADS-B-like systems, we found many similarities in terms of technologies and message protocol. Hence, we can say that Remote ID is a type of ADS-B-like formalized by the FAA into one specific means of communication for UAV.

### 3.4. Research Applications

This section will describe the research development of ADS-B, ADS-B-like communication, and Remote ID in UTM application based on the literatures categorized in Table 2.

### 3.4.1. ADS-B Research

At the beginning of UTM development, the ADS-B was considered a top candidate for UTM communication. It was supported by a regulation that ADS-B will be mandatory for all airliners by 2020. However, some disadvantages are discovered that make it unfavorable. We found that research on ADS-B for UTM focused on surveillance, detect and avoid, capacity estimation, and security.

As the main function of ADS-B in manned flight, ADS-B is used as a surveillance system in UTM. In this function, knowing the trajectory of a UAV is important. One research proposed a trajectory fusion based on an active and passive feedback system. The ADS-B surveillance data from UTM were used as the active and passive feedback. The data were derived from the ground control station. Their simulation obtained accurate flight parameters and a continuous stability of UAV trajectory [24]. Other research used a more advanced method for accurate 4D trajectory prediction: a genetic algorithm. The algorithm used historical ADS-B flight data and the UAV equation of motion. Their simulation could estimate UAV trajectory and the entering time to the protection zone accurate and instantly [25]. Since ADS-B data are heavily dependent on information from the GNSS system, another research proposed a method to construct a UAV trajectory when devoid of a GNSS signal. The algorithm used a measurement of the time elapsed from the UAV connection to several ground-based stations. Then, it used the triangulation (multi-lateration) method. The flight test showed that the algorithm was capable of navigating a UAV through several waypoints of trajectory in the missing GNSS signals [26]. Hence, the trajectory research aims to provide accurate and reliable information for surveillance functions.

The research in ADS-B as a surveillance system for UTM also covers mixed traffic between UAVs and manned flight. One of the concerns from the aviation community on the usage of ADS-B in UAVs is the frequency saturation when a high density of UAV flights becomes a reality. Two publications focused on using reduced transmission power of ADS-B to mitigate the saturation condition. First, research conducted a flight test to measure ADS-B performance in UAVs for several transmission power levels [27]. The second research simulated general aviation and UAV flights in a low-level and very low-level airspace of an urban area, using low-power ADS-B as a surveillance system [28]. Both types of research concluded that low-power ADS-B provided data at acceptable ranges and update intervals required for UTM operations. Additionally, the reduced transmission power also produced a sufficient range to alert manned flight. One more research in mixed traffic used ADS-B data to analyze the effects of lack in clock synchronization to the ATM. It provided solutions based on the availability of several trusted sensors in a large, uncoordinated network of UAVs [29]. In this context, the UAV functions as a relay airborne for ADS-B of manned flights.

In support of ADS-B as a surveillance function in UTM, several types of research also focused on designing required components, such as an antenna, ADS-B module, UAV, and airspace structure. One research designed a UAV antenna dedicated to detecting manned flights by the demodulation of ADS-B. The antenna consisted of an array of planar inverted-F antennas, a reflector plane, and a quadrature feed network. The test shows that it can detect aircrafts up to 437 km [30]. Besides antennas, another research designed and prototyped a cards system that is lightweight and small enough to be placed on a UAV, and could create and broadcast a ADS-B message compliant with RTCA DO-282B [31]. On

top of the antenna and ADS-B module, one example of UAV design dedicated for ADS-B measurement was the UAV-based flight inspection system (UFIS). This prototype UAV contained antennas, a positioning module, an airborne processing unit, a flight inspection sensor, a control unit, and an air-ground data link unit [32]. Besides vehicle components, an airspace design is also an important part of UAV surveillance using ADS-B. The SafeDrone European project designed flight procedures and an airspace related with no-fly zones to ensure safe separation of UAVs and manned flights [33]. Thus, the integration and implementation of all those designed components need to be integrated into the real UTM operation.

The second category in ADS-B research is detect and avoid. It is related to safety as the top priority in UTM operation. We found the largest amount of research focused on this area. For clarity, we categorized it into trajectory estimation, data science processing, mixed traffic detection, and system design.

For the trajectory estimation research, we divide it into two groups: algorithm and implementation. We found four publications that explained detect and avoid algorithms in UAVs using the ADS-B sensor. The first algorithm represented a method of defining a potential collision of two or more UAVs in airspace. UAV trajectories were estimated by two or three waypoints of trajectory obtained from the ADS-B system. It calculated a point of the UAVs' collision by defining the crossing points of trajectories from two cut-off values in the critical speed range [34]. Similarly, the second publication proposed a distance-limiting scheme to verify the flight distance and trajectory in the ADS-B transmitted messages from nearby UAVs [35]. Although the third algorithm used ADS-B data from nearby UAVs, it used a fuzzy logic algorithm to make a decision to avoid the collision [36]. The fourth algorithm used grey wolf optimization (GWO) to avoid moving obstacles. The position of obstacles is provided via the ADS-B or ground-based radar. Unlike the other algorithms, it assumed that the obstacles' future trajectories were unknown. The solution was calculated based on Bayesian formalism with a distance-weighting function [37]. The last algorithm in the detect and avoid category used information sharing protocol to predict the collision event. The predictive model controller is represented by a pair of UAVs coupled with the presence of an imminent collision [38].

The algorithms mentioned in the previous paragraph found that the first algorithm was successfully implemented in a flight simulation using up to 50 UAVs with yaw and speed maneuvers [39]. However, the second publication on detect and avoid implementation was not related to the mentioned algorithms. It used both software-in-the-loop and hardware-in-the-loop simulations. The UAVs used Pixhawk autopilots for autonomous flight and Intel NUC processor hardware for the collision avoidance algorithm implementation [22]. Hence, the trajectory algorithm and its implementation became critical for the successfulness of the detect and avoid function.

Besides detect and avoid techniques based on trajectory estimation, data science technology is a newly emerging method. One publication used a data science technique for the detect and avoid categories. It used data from an ADSB aggregator on six days and within 5 miles of an airport. The data showed that some zero-foot grids were well beyond the traffic pattern with no manned aircraft below 500 ft AGL for at least a mile [40]. Another publication combined ADS-B data from manned flights with UAV traffic data measured by DJI AeroScope near Daytona Beach International Airport (KDAB). The collected data were examined to define the population of UAS flights, to measure the maximum flight altitudes, and to determine the operating locations [41]. Although these data science techniques do not directly relate to detecting and avoiding UAV traffic in real-time, the data-driven analysis could be used for future planning of drone safety operations.

The detect and avoid category also considers mixed traffic between UAVs and manned flights. The most researched area in this category is the separation assessment. The NASA flight demonstration for UTM technical capability level (TCL) 3 covered a sense and avoid scenario in which it addressed the hazard posed by transponder-equipped, manned flights. The scenario demonstrated the safe lateral separation in conflicts between a UAS using ADS-B

IN and a manned flight using ADS-B OUT [42]. Similarly, another publication assessed separation between UAVs and manned flights equipped with ADS-B using commercial sensor equipment, called DJI AeroScope [43]. Unlike the previous two publications using sensors to assess the separation, one publication sought to determine the mean visibility separation of UASs in visual meteorological conditions (VMC). It aimed for an alert to a pilot flying a general aviation (GA) aircraft [44]. Besides the separation assessment conducted by three previous publications, one more publication in this category proposed a ground-based sense and avoid (GBSAA) system to enable mixed traffic of low-altitude UAVs and ADS-B-enabled, manned flights. It conducted analytical and simulation studies to investigate a possible collision due to the different network parameters. It found that the GBSAA could support more UAVs than an ADS-B-only system [45].

For supporting the detect and avoid functionality, some researchers focused on designing or implementing the required components or systems. An antenna is one of the important components. One research designed a coaxial-fed compact blade antenna with a dual frequency. It combined the ADS-B and 5G cellular network. It showed that an antenna with a low profile and a simple structure was suitable for future UAV-assisted 5G networks. Additionally, it was well suited to the upcoming ADS-B-based detection avoidance functions [46]. The second component researched was the ADS-B module for UAVs called Ping2020 from uAvionix. The publication presented the collision avoidance algorithms' implementation in a fixed-wing UAV. The simulation and flight test results showed that ADS-B's integration into UAVs was potentially effective for the detection and avoidance function [21]. Another system to support the detect and avoid function is a detection system consisting of multi-sensors. There are three publications that described this system. *The Drone Net* is one of them. The system was a combined detection system using ADS-B, RADAR/LIDAR, and an electro-optical/infra-red camera [47]. Another research proposed a combination of a common video, audio sensors, a thermal infra-red camera, and ADS-B for a potential solution to a drone detection system [48]. These two detection systems are fixed on the ground. The other detection system that can fit onboard UAVs is a lightweight obstacle detection system. It integrates a thermal infra-red (TIR) camera and ADS-B receiver [4]. All those detection systems combined ADS-B with other sensors for improvement of its robustness to sensor errors, sample loss, and false detection. The last system designed for the detect and avoid function is the drone itself. One research designed and prototyped a delivery drone equipped with an ADS-B module to improve its safety. The prototype was effective in detecting and avoiding other UAVs during their mission [49].

The third category in ADS-B research is the capacity estimation. It estimates the maximum capacity and tries to minimize the effects of limiting factors. One research demonstrated the bandwidth limitations of a 978 MHz ADS-B frequency. This limitation could be used for the feasibility study of future high-density UAV traffic in ADS-B-equipped airspace [20]. One possible usage of this limitation is in UAV flight approval. A UAV management framework called uFly was proposed to control the capacity of ADS-B-equipped UAV traffic over the air [6]. This capacity limitation applied to maintain the safety and efficiency of UTM operation.

The last category in ADS-B research is security. This is one of the main concerns in ADS-B technology adaptation to UAVs, since ADS-B protocol does not include any security features. In this category, it covers the tool for detecting security attacks and some possible solutions. The first proposed detecting tool was a probabilistic model checking (PMC) tool. The tool could model masquerading, direct access, and denial of service (DoS) attacks [50]. The second tool used spectrum-sensing algorithms to differentiate ADS-B and UAV signals. It used serial segmentation sensing multi-mode to combine ADS-B and UAV signals. Then, based on a zero-IF structure, it verified the UAV identity [51]. Once we know the type of security attack in ADS-B, we can explore the solutions. There are two types of research on the improvement of ADS-B protocols in including a security feature. The first research proposed the usage of the blockchain method as a secure authentication platform during

flight planning approval. Then, it encoded the authentication payloads in the broadcasted message during flight [52]. The second research developed an authentication method by monitoring the signal transmission time between the senders and the receivers. The actual transmission time was calculated based on a tiny timestamp value. Hence, it is called "ADS-B with timestamp" (ADS-BT) [5]. Both types of research required slight changes in ADS-B protocol to include additional security features. However, the changes in the ADS-B protocol needs an additional process for approval by the authority.

### 3.4.2. ADS-B-Like Research

The ADS-B-like communication system mimics ADS-B protocol but uses different transmission technology. So far, it is not regulated by any regulations. It comes from the research community as a suggestion to include widely available communication means to be used in UTM. We found many publications on it, which can be categorized into surveillance, detect and avoid, capacity, security, communication, and controlling categories.

The ADS-B-like system can provide position information on nearby UAVs as the surveillance function in UTM. The communication technologies used in ADS-B-like communications as surveillance systems include radio frequency link, telemetry, APRS, LoRaWAN, LTE/4G, and Link SAC. The first research developed tools to generate automated flight paths and analyze the coverage of radio frequency links along an intended flight path. It aimed to minimize the likelihood of radio frequency link loss during a flight. The tools intended to ensure that the UAV position is always monitored [53]. The second communication used for ADS-B-like surveillance is telemetry, which UAV hobbyists commonly use for manual flight. One research used telemetry to develop a geo-awareness system, in which the position of UAVs was captured from telemetry protocol [54]. The second research proposed software-defined radio transceivers combined with a machine learning algorithm to decode the UAV's telemetry protocols from the unwelcome UAVs. This software, combined with the techniques of pattern recognition, could provide an integrated system for drone surveillance [55]. The third technology in communication for ADS-B-like surveillance is LoRaWAN combined with APRS. Two publications explained this system and examined the effectiveness of UAV surveillance under 400 feet of altitude. The system was able to monitor beyond visual line-of-sight (BVLOS) operation near suburban areas with flight distances up to 8 km [16,56]. Another research for leveraging the LTE/4G network was introduced for new surveillance of UAS operations. Under the name of the vigilant UAS surveillance communication concept, it could enable air-to-air communications [8]. In military contact, one publication mentioned a communication system called link-situational awareness and control (Link-SAC) for reliable UAV control and surveillance. This system required spectrum allocation and management, because Link-SAC needs a large bandwidth under the limited spectrum resource. Thus, spectrum sharing required allocation of an X-band uplink to Link-SAC [57]. All those technologies give a different area coverage and bandwidth.

Based on the ADS-B-like surveillance technologies mentioned in the previous paragraph, some research conducted implementation in real UTM operation. The implementation of APRS systems was published in a report that consisted of 19 flight tests for surveillance. The recorded data included position and six degree-of-freedom flight data on the UTM cloud [17]. Another implementation used a multi-channel position broadcast solution based on Wi-Fi modules. This system aimed for a robust position transmission against a jamming signal [11].

The second research area in the ADS-B-like system is detect and avoid. In this area, safety is a major concern. There are several communication technologies used in detect and avoid functions, such as Wi-Fi, XBee, and DJI AeroScope. There are two publications that used Wi-Fi as a detect and avoid tool. The first publication presented a method to broadcast short messages within the service set identifier (SSID) of a Wi-Fi network. It found that the SSID was suitable for low-latency coordinate exchange in collision avoidance [58]. The second publication described the traffic management architecture and services for defining

the separation distance between UAVs to ensure safe operation. The measurement result suggested that the scheme of Wi-Fi based messaging was a potentially useful tool for the UTM [59]. The other communication type in ADS-B-like systems for the detect and avoid category is a commercial sensor called DJI AeroScope. There are two pieces of research that used this sensor. The first one was used for UAV detection near Florida's Orlando Melbourne International Airport (KMLB) [43], and the second one was used near Daytona Beach International Airport (KDAB) [41]. Both publications analyzed the recorded UAV data combined with ADS-B data from manned flights. They pointed out the potential benefit of using such technology for real-time detection applications. One more publication evaluated XBee communication as an ADS-B-like system for the detect and avoid function. The system was successfully implemented in a flight test using a detect, predict, and avoid algorithm [22]. In support of ADS-B-like usage as a detect and avoid function, a risk assessment was conducted to evaluate LoRaWAN, 4G, XBee, and APRS technologies. Using UAS logistic delivery case studies, the research conducted an assessment on the ground risk and air risk. The result mentioned acceptable data to support the UAS logistic delivery with adequate path planning [60]. However, the case study was only conducted in remote and suburban areas.

Unlike ADS-B which needs protocol changes to include a security feature, ADS-B-like communication is open for any new protocol to improve its security. We found only one publication in this area, perhaps due to the security features researched in other research fields that are not only dedicated to UAVs, such as information technology. The research proposed a stronger authentication mechanism in the 5G network for UAV communication. It was inspired from the idea of second-factor authentication, which depends on a unique drone digital identity [62]. Perhaps, other security features of the ADS-B-like system should be tested and verified.

Furthermore, one of the advantages of the ADS-B-like system is its ability to transmit payload data on its protocol. The technologies that support this feature are XBee and the 4G/5G cellular network. One publication used stereoscopic vision as a tool for sensing and detecting obstacles and other aircrafts. The video data from the ZED stereo camera were sent to the ground control station via XBee radio to be analyzed, to get the depth maps of the surroundings [61]. Another publication implemented a 4G cellular network for UAVs to transmit their location data to the cloud. Then, the ground base station retrieved the aggregated information to broadcast it to the ADS-B-enabled aircrafts via ADS-B technology [45]. This data communication link made real-time information transmission from UAVs possible.

The last research area in the ADS-B-like system is its usage as a control and command function. Since the controlling function is a critical component of UAVs, its availability is the top priority. One research used a radio frequency link as the control and command communication. It developed a tool to analyze link coverage within an intended flight path. It also worked as an automated path development tool, which generated mission plans to minimize the likelihood of radio frequency link disruption [53]. Another research on the military side introduced Link-SAC to transmit the control and command signal to UAVs. Since the Link-SAC required a large bandwidth to support its reliability, spectrum sharing was employed to allocate the required bandwidth [57]. Hence, the ADS-B-like system could support several UTM functionalities with one type of communication technology.

### 3.4.3. Remote ID Research

Remote ID protocol was introduced to identify the nearby UAV with verification based on its unique ID. It is relatively new compared to ADS-B and ADS-B-like systems; thus, fewer publications were found for this system. Besides security research, this system also attracts researchers to explore and use it for detect and avoid functions, capacity estimation, and communication.

There are three publications using Remote ID to detect and avoid. However, the technology used in each publication was different; one was LoRaWAN [12], dedicated

short-range communication (DSRC) [13], and the other was Bluetooth 5 [14]. The first publication explored Remote ID using LoRaWAN with its ground station to identify UAVs that intentionally or unintentionally fly through a restricted zone [12]. The other two publications used Remote ID as a vehicle-to-vehicle (V2V) communication. The first one used DSRC protocol to broadcast data including the position, which was used to detect and avoid UAVs [13]. The latter used Bluetooth 5 to detect and avoid between UAV and manned helicopter [14]. Although Remote ID broadcast coverage is considered smaller than ADS-B or ADS-B-like systems, it is large enough for detect and avoid functions in UTM.

The second research area in which Remote ID was used is capacity estimation. This publication explained the simulation used to assess the Bluetooth and Wi-Fi performance, to broadcast the Remote ID protocol. It found that to avoid saturation, a significant number of ground antennas will be required to support high-density traffic [63]. It is related to the small coverage area of Remote ID broadcast.

Although Remote ID was introduced for UAV security purposes, some researchers conducted research to improve and verify its security capability. Three publications suggested the improvement of UAV authentication. First, the blockchain and smart contact concepts were introduced to generate the ID and verify it using the Ethereum platform [65]. It is like the digital currency concept. Second, a privacy-preserving authentication framework was introduced that verifies the flying UAVs identity anonymously. This framework was based on the digital signature scheme of Boneh–Gentry–Lynn–Shacham (BGLS) [66]. The third publication used an embedded subscriber identification module (eSIM). The security analysis of this concept was verified using the ProVerif platform [67]. These last two concepts employed the digital signature concept to improve Remote ID security.

Moreover, the verification test of security features for Remote ID was conducted by NASA flight demonstrations in UTM TCL 3 and 4 projects. The TCL 3 project verified a Remote ID broadcast operation in which UAVs were registered with UTM public key infrastructure (PKI) and broadcasted its messages. The messages were received properly and were identified by authorities on the ground [42]. The TCL 4 project demonstrated the capability of UTM-enabled Remote ID in several scenarios. It was used to identify and contact UAV operators in the vicinity and monitor security responses of UAV flights near an airport. Additionally, it simulated a scenario in which a low-battery UAV was forced to land quickly, affecting nearby operations to re-plan and restrain the landing vehicle [68]. Based on these flight demonstrations, the FAA decided to enforce Remote ID usage in UAVs by 2023.

The last research area in the Remote ID category explored by researchers is the communication process to retrieve the verification information from the authority database. One research proposed a framework for fast retrieving UAS information based on NASA's UTM concept. The framework consisted of vehicle registration, USS (UAS service supplier), FIMS (flight information management system), and model database. The result showed that information retrieval time was 1.2 s [19]. Another publication proposed an anonymous remote identification (ARID) framework, which used ephemeral pseudonyms. In this study, only an authorized authority, such as the FAA, could retrieve the complete information of the UAV and its operator. The result showed a much faster processing time of 11.23 ms [64]. Although its result is much faster than the first publication, the framework is not well known and examined.

### 3.5. Remaining Issues

In this section, we will describe the remaining research issues that researchers can pursue in the future. The issues found in the publications will be categorized into security, safety, communication, and surveillance, as in Table 3.

### 3.5.1. Security

In the beginning, ADS-B was a promising surveillance technology for UTM. However, it encountered some issues in signal integrity and security challenges. ADS-B's main

security issues are its lack of authentication and encryption [69,70]. Since the ADS-B an inherent insecure protocol, it is vulnerable to any kind of cyber-attack [71]. Although the ADS-B transponder broadcasts the message automatically, in some incidents, it was found that the pilot can turn it off [72].

Due to the unsolved security issues in ADS-B, the ADS-B-like system, such as LTE/4G, was developed to include an extension for security protection against the security mentioned in the above issues [70]. However, the usage of unlicensed wireless communications between UAV and ground stations such as Wi-Fi are vulnerable to cyber-attacks. Moreover, many people more concerned about privacy disclosure from aerial photography taken by UAVs [75].

Unlike ADS-B and ADS-B-like systems, which have an inherent insecure protocol, Remote ID is designed to include authentication and security features approved by the FAA [73]. However, if the operator or pilot intentionally circumvent the features, the home location could be hidden, and a fake ID number could be used [72]. Additionally, the endorsement by the FAA instigated the rejection from some UAV hobbyist communities [74].

### 3.5.2. Safety

The ADS-B system may experience capacity limitations when high-density UAS traffic becomes materialized [69]. Some critiques of ADS-B usage in UAVs arise, such as that there are too many vehicles that are very close to each other, causing a lot of congestion; the communication range near the ground is much shorter; and that interference is caused for airliner pilots during take-off and landing near airports [70]. Another issue related to the insecure protocol is that authentication of detected nearby vehicles comes into question [71].

Similar to the capacity limitation in ADS-B, the ADS-B-like system requires congestion management for centralized traffic [70]. Furthermore, an ADS-B-like system such as Wi-Fi or Bluetooth as UTM communication makes V2V communication possible, especially for detection avoidance functionality [73]. Hence, more innovative solutions could be explored to improve the safety level.

Since Remote ID also employs Wi-Fi technology, V2V protocol-based detection avoidance has also become possible. However, the usage of commercial off-the-shelf devices such as LoRaWAN caused some variation in its performance [12]. Another issue experienced in the DSCR module is the initial synchronization process that took too long to establish [13]. One more consideration should be taken when involving human pilots in mixed traffic UASs with manned flights. An aircraft pilot does not easily see small UAS unless it is very close for keeping a safe distance. Although Remote ID could alert the pilot of the existence of nearby small UASs, without "in-sight" it might be difficult to decide an action for safe maneuvers without a proper alerting mechanism [14].

### 3.5.3. Communication

Since ADS-B and Remote ID are not designed for data communication, no issues were reported on this type of research. Mainly, only the ADS-B-like system can be used. In communication using LTE/4G, it has a limitation of coverage in rural areas [69]. Additionally, a redundant communication system should be considered to improve its integrity and availability [73]. One of the proposals is to include a satellite link communication [74]. Other issues in data communication are command and control or non-payload and payload communication, and whether these two types of communications could be combined or separated in their implementation [76].

### 3.5.4. Surveillance

One of the issues against the usage of ADS-B in UAV surveillance is the ADS-B price. Although ADS-B IN is relatively small and low cost, the broadcast module ADS-B OUT is expensive, making it less popular for the small UAS market [73]. Another issue is the dependency of ADS-B to the GNSS system for position accuracy [74].

For ADS-B-like systems such as LTE/4G, usage for surveillance might be improved with new improvement in 5G technology [69]. However, their coverage is still limited by the ground cellular network location. Perhaps satellite link communication could be the backup system [10].

For Remote ID, the system can share data like ADS-B for surveillance. However, to enlarge its coverage, it should be combined with other communication channels in the ADS-B-like system, or use network Remote ID via cellular network [70].

## 4. Discussion

The discussion section covers SWOT analysis and proposed UTM communication architecture based on published manuscripts found.

### 4.1. SWOT Analysis

The SWOT analysis, consisting of strength, weakness, opportunity, and threat analysis, is an effective means for strategic planning [77]. It could be used in combination with the literature review process [78]. The analysis is conducted to evaluate the available wireless communication technologies for UTM, and the result is shown in Table 4.

**Table 4.** The SWOT Analysis result.

| SWOT | ADS-B | Remote ID | ADS-B-Like |
|---|---|---|---|
| **Strength** | Mature standard and technology, surveillance purpose | Mainly for security, Established standard | Two ways of communication, many options in wireless technology |
| **Weakness** | No security feature, predicted capacity congestion | Short coverage, not supporting surveillance | No standard established, variety in quality |
| **Opportunity** | Compulsory in manned flights by civil aviation authorities | Endorsed by FAA | Wireless technology development will favor the application for UAV |
| **Threat** | Other systems have a lower price | Usage by security officers | New ground accessory required |

In ADS-B, the strength comes from the fact that it is an established surveillance system in manned flight with a well-defined standard. It is regulated to be mandatory for civil aviation in many countries. However, the lack of security features and the prediction that it will make capacity congestion become its weakness in UTM. In addition, the higher price compared to other systems make it not preferable to be adopted in the UAS.

On the other hand, Remote ID is established to be the solution for a security issue in ADS-B, with its main function being for security. It has clearly defined the standard and is endorsed by the FAA, and other countries will follow. On the other hand, the improvement in security resulted in some rejections from the public. Additionally, the short distance coverage due to Wi-Fi technology makes it unsuitable for surveillance purposes.

Although ADS-B and Remote ID were established by authorities, the ADS-B-like system has advantages in terms of technology choices and capabilities. It supports two ways of communication suitable for command and control, and payload communication. Further development of wireless technology will accommodate the UTM requirements, such as low latency and high availability. However, its disadvantages come from the fact that it used commercial, off-the-shelf products, which vary in their quality. For the surveillance purpose, a new ground accessory will be required.

### 4.2. UTM Multi-Channels Communication

Based on the finding in this review, we propose a UTM communication architecture for the UAS, which consists of multi-channels for different types of purposes, as shown in Figure 7. It aims to improve the reliability and availability of information. In general, it

consists of three types of communication links, which are vehicle-to-vehicle (V2V), vehicle-to-cloud (V2C), and stakeholders-to-cloud (S2C).

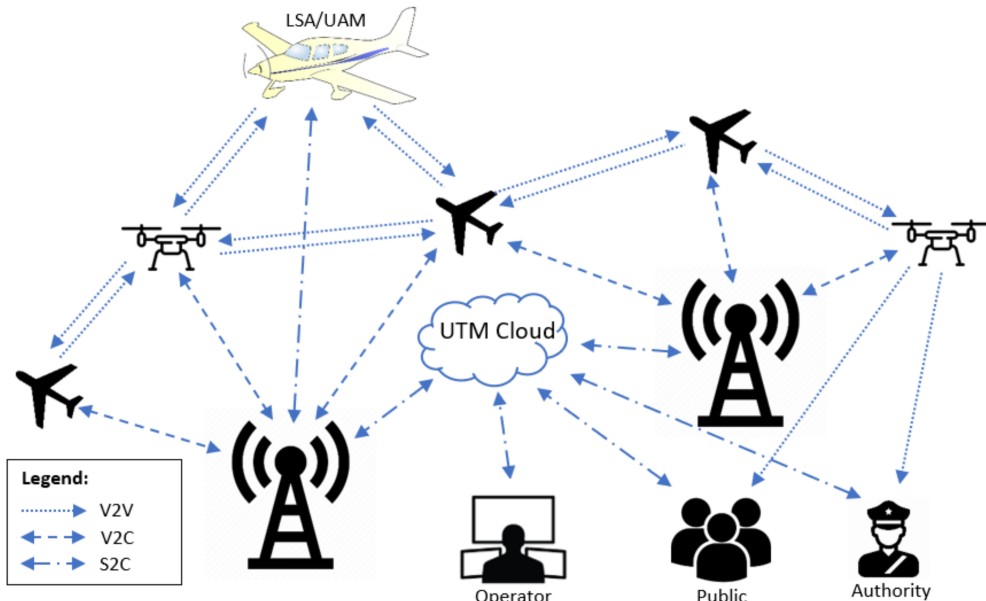

**Figure 7.** UTM multi-channel communication architecture.

V2V communication connects UAVs to other UAVs nearby. It could be used for detection and avoidance purposes in coordinated or uncoordinated mode to improve UTM safety. Broadcasted Remote ID with Wi-Fi or Bluetooth technology is the main choice due to its endorsement by regulators such as the FAA. The other choice is ADS-B, which uses radio frequency technology. However, it has some rejections due to its high price and the prediction of frequency congestion when in high-density UAV traffic, making it the second option.

The second communication link is V2C, which connects the UAV to the UTM cloud. Its capability is not only to transmit position data for surveillance, but also to command and control, and provide payload data. For this V2C link, the ADS-B-like system is the first choice. The technologies supported for this function are LoRaWAN, XBee, APRS, or 4G/5G networks. When the data transmitted are required only for surveillance purposes, network Remote ID could be used with a 4G/5G connection. For the same purpose, the usage of ADS-B is also possible.

The third communication link is the S2C network between the stakeholders and the UTM cloud server. The stakeholders of UTM are UAS operators, the public, authorities, and low-altitude, manned aircraft such as light sport aircraft (LSA) and urban air mobility (UAM). The technology for the S2C link is an internet network that could be assessed via wireless technology such as 4G/5G networks or satellite communication and the line internet network. Each of the stakeholders has its own interest and purpose for UTM. The LSA/UAM needs to be included in UTM because, in the upper part of UTM airspace, UAV could overlap with them. They are more concerned about UTM for surveillance to define their flight trajectory in the presence of UAVs around them. The second stakeholder is the public, which is concerned about UTM for surveillance and security, and needs to know what the UAVs are doing in citizens' locations. The third stakeholder is UAS operators. Besides surveillance, the UAS operators require command and control purposes and payload communication, which are suitable for their operational purpose. The last stakeholder is the authorities who are concerned about the security and safety of the whole UTM operation.

In addition, the V2V link could be used for a direct connection between UAVs and the UTM stakeholders in the vicinity. For the case of using a broadcast Remote ID, its signal can

be received by the authorized personnel to monitor the safety of UAV flights in the vicinity. It can also be used by the public for surveillance purposes. However, the stakeholders will be required to have an internet connection to the UTM cloud for data authentication. Especially for LSA/UAM stakeholders with other type of aerial vehicles, the V2V link can be used for their detect and avoid function. If they are willing, there is an option for them to adopt a similar V2V technology.

## 5. Conclusions and Recommendation

This literature review was conducted to survey the latest publications in UTM communication. It aims to give a broad overview of the available systems and suggest how to construct a feasible communication architecture. The method used is an integrated approach in reviewing and analyzing the literatures.

It concludes that the UTM communication links available today can be categorized into ADS-B, Remote ID, and ADS-B-like systems. The publications reviewed reveal that the main research applications are for surveillance, detecting and avoiding, capacity estimation, communication, command and control, and security.

From the number of publications found, ADS-B is considered as the most researched subject, and Remote ID is the least since it is the newest system. However, there are still many remaining issues that can be explored by researchers for further study. Especially in Remote ID and the combination of several communication systems, there are more issues to be explored to support of communication functions of UTM.

Based on the review, SWOT analysis is conducted to reveal the comparison of ADS-B, Remote ID, and ADS-B-like systems. Each of the systems has their advantages and disadvantages. A combination of two or all systems might complement their weakness and form stronger advantages.

The authors propose a new UTM communication architecture using multi-channel communication that could be the solution for the near-future operation of UTM. It is an improvement to the UTM-based, ADS-B-like communication architecture [16] to include ADS-B and new Remote ID technology. This multi-channel architecture could improve the UTM communication's security, availability, and reliability.

The further work from this literature review is to model and implement the proposed UTM communication architecture in simulations and flight tests. It will include the analysis of its performance and effectiveness in UTM operation.

**Author Contributions:** Conceptualization, C.-Y.L. and S.-C.C.; methodology, N.R.; formal analysis, N.R.; writing—original draft preparation, N.R.; writing—review and editing, C.-Y.L. and S.-C.C. All authors have read and agreed to the published version of the manuscript.

**Funding:** This research received no external funding.

**Conflicts of Interest:** The authors declare no conflict of interest.

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
