# Peer review of "UAS Traffic Management Communications: The Legacy of ADS-B, New Establishment of Remote ID, or Leverage of ADS-B-Like Systems?"

_drones, doi:10.3390/drones6030057_

Round 1
Reviewer 1 Report
Introduction: Please cite forecasted growth.
Table 22, does it depicts that there were no paper in the vacant place?
Figure 3.1 is not clear, please make a vector image; same goes for 3.2 and 3.3
Conclusion section is missing in the paper. Summary can be further extended.
References are too less being a survey paper.
Novelty is missing, i would suggest to create a SOTA table and differentiate your work.
Readability is missing in the paper as there are too frequent context switching and paragraph change.
Future work and open ended research directions should be discussed in summary section.
Overall, paper is good but miss to clearly mention novelty and give future research directions along with missing conclusion.
Author Response
- Introduction: Please cite forecasted growth. Answer: updated in line 28-33
- Table 2.2, does it depict that there were no paper in the vacant place? Answer: Yes, it does. For example: no paper on ADS-B for communication, and command and control because logically ADS-B not capable for communication and command and control.
- Figure 3.1 is not clear please make a vector image, same goes for 3.2 and 3.3. Answer: updated images for Figure 3.1, 3.2 and 3.3.
- Conclusion section is missing in the paper. Summary can be further extended. Answer: Section 5 become Conclusion and Recommendation. The content also updated to include the proposed UTM multi-channels communication architecture.
- References are too less being a survey paper. Answer: References are enlarged to include 72 published conference papers and journals. It is mentioned in line 86-100.
- Novelty is missing, I would suggest to create a SOTA table and differentiate your work. Answer: Novelty mentioning is added in line 54-58. Major changes in sections 3.4 and 3.5 to focus on current research development and remaining issues for future direction. Also new section added in 4.2 for proposed new UTM multi-channels communication architecture.
- Readability is missing in the paper as there are too frequent context switching and paragraph change. Answer: I am sorry for my English. I try my best to smoothing the paragraph flow for more readable text.
- Future work and open ended research directions should be discussed in summary section. Answer: Future work now is mentioned in line 729-736. It focuses on the new proposed communication architecture.

Reviewer 2 Report
In the introduction, originality must be justified in the context of previous studies. The author should update the manuscript with appropriate and relevant research questions. The importance of research does not arise. Results presentation section seems to be general. The conclusions section must be improved clearly indicating the main conclusion, and relating it to the literature supporting the paper ("according to (...), or" unlike (...) "). Also, conclusions should be rewritten to understand the importance of research. Are there any further research streams?
Author Response
- In the introduction, originality must be justified in the context of previous studies. Answer: Novelty mentioning is added in line 54-58.
- The author should update the manuscript with appropriate and relevant research questions. Answer: References are enlarged to include 72 published conference papers and journals. It is mentioned in line 86-100.
- The importance of research does not arise. Answer: A new section added in 4.2 for proposed new UTM multi-channels communication architecture. This is the major contribution from this review.
- Result presentation section seems to be general. Answer: Major changes in sections 3.4 and 3.5 to focus on current research development and remaining issues for future direction.
- The conclusions section must be improved clearly indicating the main conclusion and relating it to the literature supporting the paper. Also, conclusions should be rewritten to understand the importance of research. Answer: Section 5 become Conclusion and Recommendation. The content also updated to include the proposed UTM multi-channels communication architecture.
- Are there any future research streams? Answer: Yes, there are. Future work now is mentioned in line 729-736. It focuses on the new proposed communication architecture.

Reviewer 3 Report
The paper presents a review of the literature on three different communication protocols between Unmanned Aerial Systems (UAS), namely ADS-B, Remote ID, and ADS-B-like. It analyzes the protocols from five perspectives: definition, format, technology, application, and issues.
I find that the review is not of sufficient quality to be published due to, among others, the following reasons:
1.- The survey is based on a rather small and somewhat biased set of papers. The authors searched for a few terms in Google Scholar and then took the top results published during the last 5 years for which the full text was available. There was no control for the quality of such articles, number of citations, source, etc.
2.- The paper includes many technical details which seem irrelevant to the discussion, such as the frame structure in Figure 3.3. or the mean message rates. If they have some effect on the performance of the protocols, the authors should point it out. Otherwise, the paper should focus on the different options that have been proposed rather than on the underlying standard that all of them are based on.
3.- I do not see this review being very useful. It has lots of lists, but little description or analysis of the items. For example, in page 7 lines 220-225 it describes many possible technologies for ADS-B Like systems, but it does not say what it means by "technologies", neither what the differences between them are. In general, the review is very superficial.
4.- The paper has lots of figures, which can be a good thing when used to clarify the content of the paper, but in this case they often cause additional confusion introducing terms and blocks not mentioned in the paper. Furthermore, it seems that many of the figures have been copy-pasted from the references, such as Figure 3.7 from reference [7].
5.- I am not an expert in these protocols, but I am suspicious of the fact that a literature review of three different communication systems has only 26 references.
6.- The organization and writing of the paper is confusing, with many syntax mistakes that hamper the understanding. Extensive editing is needed.
Author Response
Here is my answers to your feedbaks:
- References are enlarged to include 72 published conference papers and journals. It is mentioned in line 86-100. I used several search engines to get the literatures and filtered only for peer reviewed literatures such as conference proceedings or journals.
- Major changes in sections 3.4 and 3.5 to focus on current research development and remaining issues for future direction based on more literatures.
- A new section added in 4.2 for proposed new UTM multi-channels communication architecture. This is the major contribution from this review.
- Figures in section 3.4 and 3.5 are removed to avoid distraction on the text
- The categorization of UTM communication is 3, because beside ADS-B and Remote ID, other communication channels can be categorized as ADS-B Like.
- I am sorry for my English. I try my best to organize the paragraph and write of the paper in English to be readable for everyone.
Hopefully, you are satisfied with my revision.

Round 2
Reviewer 3 Report
The authors have addressed my comment about having too few references and added a significant number of new ones. As a matter of fact, the length of the paper has pretty much doubled due to the increased length of sections 3.4 and 3.5, where they discuss the new references.
As a result, I find that the paper could now be useful to other readers as a review paper. That said, my concerns on the writing and the organization still remain. The paper has many grammar and syntax mistakes, it requires significant editing work (which the journal editorial team might undertake, I do not know how this journal works). The organization could also be improved.